# Pathways to service access for pre-eclampsia and eclampsia in rural Bangladesh: Exploring women's care-seeking

**Amy Dempsey**[1]*, **Pooja Sripad**[1], **Kanij Sultana**[2], **Karen Kirk**[1], **Sharif Mohammed Ismail Hossain**[2], **Charlotte Warren**[1]

**1** Population Council, Washington, DC, United States of America, **2** Population Council, Dhaka, Bangladesh

* amyjade13@gmail.com

## Abstract

### Background

While women in low- and middle-income countries face a range of barriers to accessing care for hypertensive disorders of pregnancy, there is little understanding of the pathways taken to overcome these constraints and reach the services they need. This study explores the perspectives of women and communities on the influences that impact care-seeking decisions and pathways to health services.

### Methods

To understand individual perspectives, we conducted 22 in-depth interviews (IDIs) with pre-eclampsia and eclampsia survivors (PE/E) in a tertiary hospital, where they received care after initiating PE/E services in different parts of the country. In four districts, we conducted one male and one female focus group discussion (FGD) to unearth care-seeking pathways and explore normative perspectives and the range of internal and external influences. Careful thematic analysis using Atlas-ti was applied.

### Results

Prevailing views of women and communities across settings in Bangladesh indicate varied pathways to care throughout their pregnancy, during childbirth, and in the postnatal period influenced by internal and external factors at the individual, familial, social, and health systems levels. Internal influences draw on women's own awareness of hypertension complications and options, and their ability to decide to seek care. External factors include social influences like family and community norms, culturally-accepted alternatives, and community perceptions of the health system's capacity to provide quality care. The interaction of these factors often delay care seeking and can lead to complex pathways to care.

### Conclusion

Women's individual pathways to care were diverse, despite the homogenous community perceptions of the influences on women's care-seeking behaviors. This finding supports the

**Data Availability Statement:** There are institutional legal restrictions on sharing de-identified data sets, however data is available on request. Request maybe sent to Population Council, Dataverse,

email; publications@popcouncil.org for information on data access.

**Funding:** This manuscript is made possible by the generous support of the American people through the United States Agency for International Development (USAID) under the terms of USAID APS-OAA-14-000048. https://www.usaid.gov/ The funders had no role in study design, data collection and analysis, decision to publish, or preparation of the manuscript.

**Competing interests:** The authors have declared that no competing interests exist.

need for improving quality of care in primary healthcare facilities and strengthening gender equity and community-based promotion activities through targeted policy and programming.

## Introduction

The ways in which women access pregnancy-related healthcare services, including services for complications like pre-eclampsia and eclampsia (PE/E), has garnered increasing attention globally. In Bangladesh, the majority of births and related complications occur in the home, without skilled birth attendants. When symptoms of PE/E occur, women may not immediately seek care through the formal health system due to a lack of knowledge of complications, preparedness, decision-making power, and real or perceived poor-quality care at healthcare facilities [1]. We sought to understand the unique perspectives of women's complex pathways to healthcare by eliciting the perspectives of PE/E survivors in rural Bangladesh.

Hypertensive disorders of pregnancy (HDPs), specifically PE/E, increasingly contribute to global maternal and newborn mortality and morbidity. Their etiology and manifestations are often misunderstood, which challenges communities' and health systems' ability to effectively prevent adverse outcomes. As countries' maternal mortality burdens decrease, non-communicable diseases (NCDs), including HDPs, comprise a larger portion of pregnancy-related deaths and morbidities [2]. Demographic transitions, including changes in lifestyle and diet, that contribute to the burden of NCDs, as well as the management of other causes of maternal mortality—postpartum hemorrhage and sepsis—compound the difficulties that health systems [3] face in preventing, detecting and treating HDPs.

In addition to clinical and health systems issues that negatively impact pregnant women and their families, deaths from PE/E occur largely as a result of delays in seeking health services [4] and the use of one or more health systems [5, 6]. Thaddeus and Maine posit that delays in seeking care can occur at three different points: [1] delay in deciding to seek care, [2] delay in arriving at a health facility, and [3] delay in receiving quality care at a facility [7]. This paper focuses primarily on the first and second delays, and specifically addresses the influences that impact women's decision-making processes and their ability to receive timely care at a facility. These influences include inadequate awareness of the signs and symptoms of pregnancy-related complications, community perceptions that facility providers are unskilled and unable to manage pregnancy complications and that facilities are not equipped with essential drugs and commodities [8, 9].

In South Asia, and specifically Bangladesh, PE/E-related mortality is a major concern, despite strides in reducing child and infant mortality and increased contraceptive use. Each year in Bangladesh, between 5,000 and 6,000 women die during pregnancy or childbirth, and up to 1,200 –or 24 percent–of those deaths are from eclampsia [10], the second most common cause of maternal mortality after postpartum hemorrhage. There is no single barrier to accessing maternal health services, but rather multiple barriers that include social barriers within communities, as well as barriers within individual households and health systems. Each of these play a role in women's ability to access life-saving care.

In rural settings in Bangladesh, women seek healthcare services at the facilities nearest to their homes, but these facilities may not have the capacity to provide comprehensive antenatal and postnatal care [11], which includes looking for PE/E danger signs and complications, such as high blood pressure, urine albumin, swollen hands and feet, severe headaches/dizziness, blurring of vision or convulsions in pregnant women. Studies in Bangladesh have focused on risk factors, health outcomes, and interventions related to PE/E, and many found a lack of

awareness [12] among pregnant women and their families about when to seek care as pregnancy-related complications arise. While studies of maternal complications have often either focused on specific regions in Bangladesh or covered nationally representative surveys, few studies have explored the lived experiences of HDPs in geographically spread districts in Bangladesh. However, the majority of the literature focuses on urban settings, with very little focusing on the experience of rural women, their families, or providers at primary facilities.

These layers of complexity deserve deeper examination, with an in-depth look at the perspectives and motivations of women and their families. The objective of this study was to better understand women's and communities' perspectives on the influences that impact their care-seeking decisions and pathways to health services.

## Methods

This study is part of a five-country research initiative that examines PE/E in low-resource settings and investigates policy-, health system-, and community-level impacts on access to care. It aims to understand community perceptions of the disorder and the ways in which those perceptions impact women's pathways to care when they experience pregnancy-related complications, specifically those due to PE/E. The study adopted a cross-sectional qualitative approach best suited for exploring in-depth perspectives of individual narratives that unearth the complexities of care-seeking pathways and and community views on pathways to accessing care in the Bangladeshi context. We conducted in-depth interviews (IDIs) with PE/E survivors and focus group discussions (FGDs) with married men and married women to better understand perceptions around PE/E, including knowledge about the condition and the barriers and enablers of accessing antenatal and delivery care. The IDIs and FGDs consisted of open-ended and exploratory questions, with researchers asking probing questions for more specific and detailed responses.

### Study setting

Bangladesh is a homogenous country in terms of culture, language, cultural norms, education, health facility and ethnicity. The study was conducted in four districts and is part of the larger *Ending Eclampsia* project, which aimed to identify barriers to PE/E care in five countries. The four districts were Tangail, which has a population density of 975 people per square kilometer; Bhola, which has a population density of 456 people per square kilometer; Rangpur, which has a population density of 1,101 people per square kilometer; and Cumilla, which has a population density of 1,290 people per square kilometer. These districts were chosen after consultation with the Ministry of Health. Cumilla is southeast of Dhaka and consists of 16 upazilas, with most people engaged in agriculture, trade and textile work. Tangail consists of 12 upazilas. With close proximity to Dhaka, the urban growth rate in these two districts is increasing. With eight upazilas, Rangpur is in northern Bangladesh and Bhola is an island located in the south-central part of the country and has seven upazilas.

There are four main tiers within the Bangladeshi health system structure: Medical colleges and specialized hospitals; secondary facilities that provide medical and surgical services and comprehensive emergency obstetric and newborn care (CEmONC); primary facilities designed for primary healthcare, including antenatal (ANC), postnatal (PNC) and basic emergency obstetric and neonatal care (BEmONC); and community clinics that provide general health services, ANC and PNC, and short acting contraception. See Table 1 for more details.

### Study participants

Participants were from a convenience sample selected based on their lived experiences of maternal complications and diversity in care-seeking pathways in rural Bangladesh. All

**Table 1. Levels, types of facilities and providers, and responsibilities assigned to four health system tiers in Bangladesh.**

| Level of care | Type of facility | Type of providers | Services provided |
|---|---|---|---|
| Tertiary facilities (68 in total) | Medical colleges and specialized hospitals | Medical and surgical specialist including OB/GYN specialists, medical officers | Medical and surgical services, including CEmOC, and mother and child health services, including ANC, PNC, normal delivery and caesarian section, and all family planning (FP) services, vaccination |
| Secondary facilities (632 in total) | District hospitals (DH); Mother and Child Welfare Centers (MCWC); Upazila health complex (UHC) | Doctors, medical officers, nurses, and family welfare visitors (FWV) | Medical and surgical services, including CEmOC, and mother and child health services, including normal delivery and caesarian section, and all family planning (FP) services; Normal delivery, caesarian sections and family planning services; General and minor surgical services including FP, upgraded UHCs also provide CEmONC |
| Primary Health Care Level (3,942 in total) | Union and health and family welfare centers (UH&FWCs) | Family Welfare Visitors (FWV), Sub-assistant Community Medical Officer (SACMO)/ Medical Assistants | Antenatal and postnatal care, short acting contraception and child health services. Upgraded UH&FWCs provide normal delivery, basic EmONC, long acting and reversible contraception and permanent contraception |
| Community Level (13,235 in total) | Community clinics (CCs) | Community health care provider (CHCP), Community skilled birth attendants (CSBA), Family Welfare Assistants (FWA), Health Assistant | General health services, antenatal and postnatal care, and short acting contraception |

participants were recruited from tertiary hospitals, where they received care after initiating PE/E services in different regions of Bangladesh.

To understand how individual women experienced PE/E and related care, researchers conducted IDIs with 22 consenting women who had PE/E and survived with a living newborn. Our sample of survivors were between 18 and 35 years old. Half reported becoming pregnant between 18 and 20 years of age, while five became pregnant when they were younger than 18. About one-third were illiterate, while the remaining two-thirds had some education.

During the IDIs, there were two examples where the survivor could not recall the details of her PE/E complications, given the unconscious state she was in during that time. A companion who was present when the survivor experienced those complications and was present during the IDI communicated to researchers the missing details of the survivor's pathway and experience.

To elicit community perspectives on knowledge, beliefs, and behaviors related to pregnancy and childbirth, researchers conducted eight FGDs (7–9 persons per group) with married men (age 18–55 years) and with married women (18–49 years) who had at least one living child. One female and one male FGD were conducted in each district.

## Data collection

Before beginning the study, approval and administrative orders were collected from Directorate General of Family Planning (DGFP) and Directorate General of Health Services (DGHS). Hospital ethical approval was obtained from the hospital ethical committee. The data collection team was recruited and trained on the study objectives, tools, procedures, and research ethics in dealing with human subjects. When possible, the data collectors were women with prior experience interviewing women on sensitive issues. Participants received information on informed consent procedures before the data collectors conducted interviews. The IDI and FGD guides are attached as appendices.

## Analysis

Researchers analyzed data according to rigorous methods and used a thematic analysis approach that drew on emergent themes and the research questions [13]. Interviews were

digitally audio recorded and supported by hand-written notes. Transcripts were transcribed verbatim and translated into English for content. An inductively derived codebook was developed after reading the IDI and FGD transcripts and generating themes based on those transcripts. One team member applied a codebook using Atlas-ti. The team discussed the outputs of these codes, which were the catalyst for this manuscript about the pathways to care from individual and group-normative perspectives.

Coders had a social science or public health background, were intimately involved in the *Ending Eclampsia* project, and were very familiar with the concepts of PE/E research. Influences on care seeking pathways emerged as areas of interest. In this paper, we describe two types of influences: 1) Internal influences defined as a woman making a care-seeking decision based on her own knowledge and ability, and without consultation or advice from her family, neighbors or community; and 2) External influences defined as care-seeking decisions made based on consultation and/or advice from the family, neighbors or community, as well as perceptions of whether the health system can provide quality care.

Preliminary outputs and themes were confirmed and refined during in-country dissemination meetings with critical maternal health stakeholders in Bangladesh, specifically within the PE/E sector. This served as an initial member-checking step to ensure credibility of the data.

### Ethics review

The study protocol P693 was approved by the Population Council IRB in New York, the Bangladesh Medical Research Council (BMRC) and the Ethical Committee of Dhaka Medical College.

## Findings

Findings show that women in rural Bangladesh follow complex pathways as they access maternal health services, often making multiple informal and formal contacts before receiving the care they need. In many instances, these women bypassed the formal health system altogether, but also made a combination of contacts with the formal health system that span across the antenatal, delivery, and postnatal periods and often involve pregnant women traveling between multiple facilities and providers. Internal and external influences for each contact with the health system are described through the perspectives of individual PE/E survivors and community men and women.

### Pathways to accessing care

Of the 22 women surveyed, all of them had contact with the formal health system at least once during their pregnancies or in the postnatal period, and most contacts occurred at health facilities at the onset of complications. All women made their last contact with the health system at a tertiary hospital, where they received treatment in the eclampsia ward. Twenty-one women attended at least one ANC visit, while one woman did not receive any ANC, but went to a tertiary facility for a sonogram to know the sex of her baby.

The PE/E survivors interviewed had between one and 11 contacts with a healthcare facility throughout the pregnancy continuum, with an average of five contacts with the health system. During the antenatal period, women exhibited varied health-seeking behaviors; however, typically they first sought advice from family members and friends or informal providers, then traveled to primary facilities (seeing either a Sub-assistant Community Medical Officer (SACMO) or Family Welfare Assistants (FWV)) before ending up at a tertiary district hospital. The women who received care at or after the onset of complications typically went first to primary facilities and then tertiary hospitals. There were five postnatal PE/E cases among women

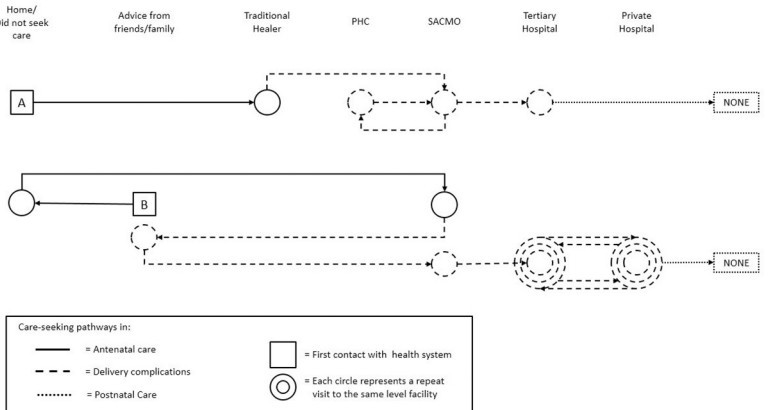

**Fig 1. Women without ANC and their unique pathways to pre-eclampsia care.**

who experienced complications during the postnatal period and had previously sought care from SACMOs at primary facilities, or at tertiary hospitals or private facilities, but again had contact with FWVs at primary facilities or private facilities at the onset of symptoms before ending at a tertiary hospital.

In Figs 1, 2 and 3 below, each individual boxed letter–A, B, C, D, E, F, and G–represents an individual woman and where she initiated her pathway to care. Each woman's pathway to care is depicted by solid (ANC), dashed (delivery care), or dotted (PNC) lines that indicate when she initiated her next contact with formal or informal health systems. Concentric circles indicate a woman's repeat visits to the same facility.

These examples show the full range of women's care-seeking experiences. These pathways varied and showed that there is not one typical pathway to care. Rather, they involved combinations of staying home without seeking care, seeking advice from family and friends, visiting traditional healers, and making contact with the health system at different points in their pregnancies.

Fig 1 shows one woman (A), who did not seek any ANC, but visited a traditional healer when she experienced complications during delivery. As she experienced complications, she then traveled by car to a primary facility, where she received care from a SACMO, and then went to another PHC and then back to the SACMO. She then went to a tertiary hospital by car for delivery, and after being discharged from the hospital, she did not seek any additional care in the postnatal period. Another woman (B) initially chose to forego ANC at a primary

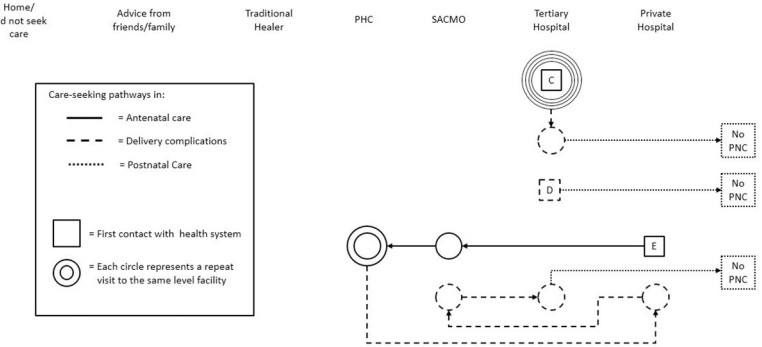

**Fig 2. Women who experience complications during delivery and their unique pathways to pre-eclampsia care.**

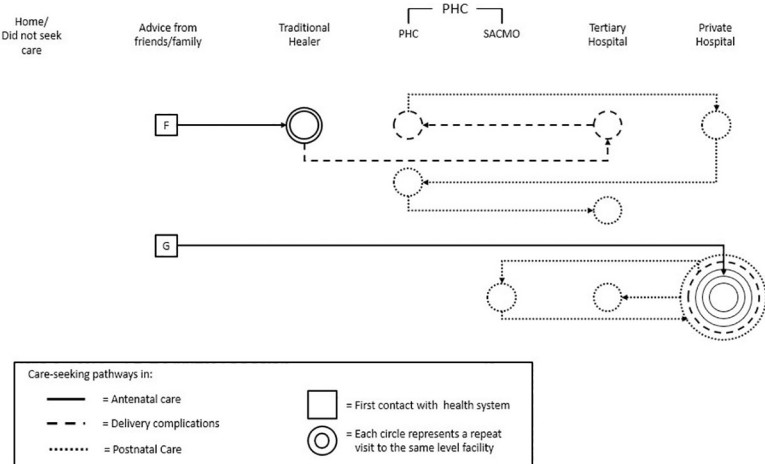

**Fig 3. Women who experienced complications after delivery and their unique pathways to pre-eclampsia care.**

healthcare facility and instead sought advice from family and friends, but later traveled by car to a SACMO at a primary facility for ANC. The same woman experienced complications during delivery, and again sought advice from family and friends, then visited a SACMO at a primary facility, and from there went between the tertiary facility and a private hospital until she had a total of 11 points of contact with the health system. She delivered her baby at a tertiary hospital, and traveled there by ambulance. Neither of these women attended PNC. There was no difference in ways in which these pathways arose in the four districts.

Fig 2 shows women C, D, and E. Woman C had five ANC contacts with the tertiary hospital, where she also delivered her baby. Woman D started her pathway to PE/E care by traveling to a tertiary hospital after experiencing complications during delivery. Woman E attended ANC at a private hospital first, then went to a SACMO at primary facility once, and to an FWV at primary facility twice. During delivery, she experienced complications and traveled back to the private hospital where she attended ANC, then she went back to a SACMO at a primary facility and then to a tertiary facility. None of these women attended any PNC, and they traveled to these health facilities by car or by bus. These pathways were seen across the four districts.

In Fig 3, Woman F first sought advice from friends and family, then visited a traditional healer twice, without receiving any ANC. She then traveled in a rickshaw to a tertiary hospital, then to an FWV at a primary facility, then to a private hospital. She then went back to the primary facility, and from there was taken to a tertiary hospital. Woman G first sought advice from family and friends, then went for two ANC visits at a private hospital and then experienced complications related to PE/E during delivery, when she went to a tertiary facility and then to a primary facility, where she delivered her baby. After delivery, she again experienced complications and bounced between private, primary, and tertiary facilities for postnatal complications. She had a total of eight points of contact with health facilities. These pathways were the same across the four districts.

## Internal influence: Maternal care-seeking

Respondents describe several factors that impact women's willingness and ability to access health services during pregnancy, delivery, and the postnatal period. However, a woman's ability to access services depends on cultural norms around childbirth and awareness of the

importance of ANC and PNC. Some respondents reported traveling to facilities only to arrive and be told there was insufficient equipment and commodities at the facilities.

> *"Economically solvent people go to the private clinics, as the treatment management is not good in the upazila hospital."* FGD, Male

> *"There is no arrangement of measuring blood pressure in our hospital. We go to the upazila hospital."* FGD, Male

> *"Her eyes were rolling and she was biting her lips! And the seizures started. We were told to take her to Dhaka Medical College, otherwise she would not survive because [the facility] didn't have any medicines."*–IDI, Female

A delay commonly reported by women regarding visiting a formal health facility was a lack of awareness of the importance of ANC or potential complications during pregnancy, specifically the signs, symptoms, and possible outcomes of PE/E. Women were more knowledgeable than men, as men considered pregnancy and delivery to be women's issues. Women describe their mothers/in-laws, female relatives, and neighbors as primary sources of pregnancy- and childbirth-related information.

> *"We have never seen or heard of such an illness. Before having the convulsion, I wasn't feeling well. I started vomiting continuously and I had stomach burning. I tried to sleep after that."* IDI, Eclampsia survivor

> *"When we got out of the house and hopped on a rickshaw, that is when the convulsion started. I didn't understand what was happening. I and a neighbor were taking her to the hospital. The doctor then said that she needs to get admitted for prescription medicine. If she doesn't get admitted, she'll die."* IDI, Neighbor of eclampsia survivor

> *"How would she know, even when I haven't heard or seen anything like this [convulsion]? I have never seen anyone having seizures."* IDI, Mother of eclampsia survivor

Women who attended ANC visits were rarely informed by health providers about potential complications or the importance of blood pressure measurement or screening for proteinuria–two methods used to detect PE/E. A few PE/E survivors reported that they were aware pregnant women could have convulsions from high blood pressure, two read this information on a poster in the tertiary hospital, one heard by word-of-mouth, one learned from service providers at a BRAC facility, and one knew from the service card she received at ANC from a tertiary hospital.

## External influence: Social influences on care-seeking during childbirth

In many cases, women are not empowered to make their own decisions about seeking and receiving health services, even during pregnancy. The FGDs showed that these decisions are often made by husbands, as they are often the sole financial contributor to a family, and thus the sole decision maker in a family. The FGDs also describe religious leaders as trusted, respected members of their communities, making them key influencers of women's health-seeking behaviors during pregnancy. After marriage, it is common for a woman to move into her husband's parents' home, and as a result, mothers-in-law also influence women's health-seeking practices. FGDs also suggest that socio-cultural influences may lead women and communities to attribute symptoms like convulsions to witchcraft, a pregnant woman's concern about the baby's gender, or God's will, and may prompt women to visit traditional and herbal

healers before seeking care at a formal health facility. FGD participants perceived signs and symptoms of PE/E, such as headaches, blurred vision, dizziness, and swelling of hands and feet as normal during pregnancy, and instead of encouraging a pregnant woman to visit a health facility, they often suggest traditional remedies, such as herbs, exorcism, charmed amulets, or holy water.

> *"These are women's problems during pregnancy and after delivery. My mother also passed through these. So, I don't want to interfere with women's problems and step in. My mother and grandmother might be the right people to decide."* FGD, Male

> "*My husband intends to take me to the hospital, but the mothers-in-law say that nowadays daughters and daughters-in-law don't have faith in Allah. They say that when they gave birth they did not go to the doctor or hospital, so they don't want to take them to the hospital. But, they may take the woman to the doctor when it is too late and mother and child's condition has become serious."* FGD, Female

> *"They [religious leaders] believe in 'holy water,' herbs, exorcism, and charmed amulets."* FGD, Female

Five of the women first sought advice from family and friends before visiting a formal health facility for ANC. Of those five women, two did not seek any formal ANC from skilled providers, but experienced complications during delivery, and at that time received medical attention from the formal health system. Those who did not seek advice from family or friends attended between one and five ANC visits at various primary, tertiary, and private hospitals. Women often consult family and/or friends first and rarely visit the PHC level, especially if they have little experience of ANC from previous pregnancies or if their family and peers did not attend ANC.

At the onset of complications during delivery, four PE/E survivors first sought advice from family and friends, while the other 18 sought services at formal public and private facilities at the onset of complications. Of those 18 PE/E survivors, five experienced postnatal PE/E.

## External influence: Health system influences on maternal care seeking–provision

External influences on maternal care seeking reported by FGD respondents reflect actual experiences of, and perceived views on, the quality of the health system and of health providers' capacity to attend to pregnant women and manage complications if/when they arise.

> *"They come to this hospital, but they cannot know their blood pressure, as there is no arrangement of measuring blood pressure and no arrangement of running any tests."* FGD, Male

> *"We go to the village doctors or Kobiraj [traditional or spiritual healer] seeking health services because of finances and lack of good treatment at the hospital. There is no good doctor in our hospital."* FGD, Female

Those who interacted with the health system during the antenatal period exhibited varied health-seeking behaviors; however, they typically first sought advice from family members and friends, then traveled to multiple primary facilities before ending up at a tertiary district hospital. The women who received care at or after the onset of complications around delivery typically went first to see SACMOs at primary facilities and then to tertiary hospitals. Women who experienced complications during the postnatal period had previously sought care from

SACMOs at primary facilities, tertiary hospitals, or private facilities, but again visited primary facilities or private facilities at the onset of symptoms before ending at a tertiary hospital.

In the IDIs, women reported their experiences with the health system as barriers to seeking care. Respondents mentioned an insufficient number of providers at primary facilities, low-quality care, and providers' inability to manage emergency situations. They also discussed inadequate referral systems from primary to secondary or tertiary facilities as a disincentive for accessing health services.

*"Behavior of the service provider at the private hospital where delivery was conducted was very bad. They left the baby on the chair. Nobody instructed us to feed the baby through the night. . .After my daughter had a convulsion. . .we got her into the ambulance, but they detained us and asked for the 32000 taka to pay the hospital bill."* IDI, Mother of eclampsia survivor

*"There was no doctor. The gateman didn't want to allow us inside. I asked him where the doctor was. . . He answered that the day after was a holiday. So, there was no doctor. We didn't get a chance at the general hospital. After that we went to a private clinic."* IDI, Eclampsia survivor

## Discussion

Prevailing views of women and communities in rural Bangladesh indicate varied pathways to care throughout their pregnancy, during childbirth, and in the postnatal period influenced by internal and external factors at the individual, familial, social, and health systems levels. Internal factors, such as women's awareness of relevant complications and options with respect to hypertension and PE/E in particular, as well as their financial means and autonomous ability to make a decision to seek care, were often modified by external factors. Socio-cultural norms around restricted female mobility, childbirth-oriented practices of hiding one's pregnancy, and the role of spouses, mothers-in-law, and neighbors somewhat hindered women's access to ANC, but often were overcome when a woman experienced more visible complications. Respondents lived in rural settings, and similar to other research [14] that examined the impact rurality has on maternal health outcomes, this influenced their pathways to care and the ways in which they traveled to health facilities, including diverse transportation methods that ranged from cars to rickshaws to buses. Health systems deficiencies, including provider absenteeism at primary healthcare facilities, compound these delays and often lead to variable pathways to care [9].

Women self-report dynamic care-seeking couched in an existing pluralistic health system in Bangladesh, where public-private and traditional avenues exist as options. These mixed care-seeking pathways can delay access to life-saving PE/E care in some cases, while in others supply and equipment limitations of PHCs reveal reasonable choices, given a local knowledge of health systems factors. Survivors frequently bypass primary health facilities or see FWVs at these facilities, either due to the perception that they offer inadequate quality care or because SACMOs at primary or tertiary facilities are more geographically accessible and offer better services [11, 13]. This was exhibited by our PE/E survivor "B," who sought ANC and delivery care at a tertiary medical college hospital. Our study concurs with prior research in Bangladesh that shows similar patterns of care seeking for NCDs in northwest Bangladesh, where non-certified providers usually failed to treat chronic conditions and women and their families sought treatment from the formal health system for advanced stages of disease [15].

In contrasting individual experiences and self-reported pathways to care with normative perceptions around care-seeking, our study found that internal and external influences do not

always align and perceived community norms may not reflect individual behaviors. Normative perceptions reveal that communities assume pregnant women avoid contacts with the formal health system and seek care and guidance from village doctors, Imams, or mothers-in-law, or do not seek care at all. Similarities among community men and women indicate that husbands are the primary decision makers, while in some cases, mothers-in-law wield more power around pregnancy-related care seeking. In contrast, individual perspectives of women who had complications suggest some autonomy in seeking care through the formal health system, in contrast to traditional sources of care. Similar to findings from other studies, these women continue to express that external influences play a supportive or constraining role in facilitating access to care [8].

Similar to other studies, our sample of PE/E survivors was composed of younger women between 18 and 35 years old. Half of the sample reported their first pregnancy occurred between 18 and 20 years old, and five first became pregnant before they were 18 years of age. Survivors who were pregnant and under the age of 18 years describe lack of awareness of the importance of ANC and complications in pregnancy as influential internal factors, but being accompanied by a spouse or older female family member and deficiencies in quality of ANC as salient external influences that affected their pathway to service access. About one-third of the sample had low education (5th grade and less); these women faced health systems and poverty barriers, including a lacking receipt (or comprehension) of provider explanations of their blood pressure and urine test results, as well as financial constraints. Our findings align with other studies that examined maternal healthcare seeking behavior among adolescent girls in Bangladesh, and suggest that women's lack of knowledge about the importance of ANC and its signs and symptoms may negatively interact with socio-economic status of women [16, 17] and affect internal processing of decisions among younger women [14].

Key policy/program recommendations that emerge out of these findings are similar to research findings from Nigeria [18] and center around the importance of bringing health services closer to the people who need them. Enabling earlier ANC use and empowering women and their families to identify signs and symptoms of PE/E and other maternal complications and seek care will minimize the risk of adverse maternal and newborn outcomes. This could be implemented through increased and targeted educational strategies at the community level. Non-distinct patterning of survivor perspectives across Tangail, Bhola, Rangpur, and Cumilla districts suggests that similar strategic programming may be applied and scaled in relatively homogenous populations in rural Bangladesh. Moreover, the larger gap in community knowledge merits attention of program targets and implies the need to strengthen the role of community health systems in maternal healthcare.

There are intergenerational differences between younger and older women and their perceptions of the formal health system and beliefs about seeking formal health services, particularly during pregnancy and childbirth. This significantly influences how families support women's care-seeking and deserves further investigation. Additionally, community awareness building around hypertension in pregnancy, dispelling myths around childbirth complications, working with faith leaders to change perceptions of the formal health system, emphasizing more equitable healthcare-seeking decisions among male-partners and joint-family members in the home are paramount.

Limitations of this study include our inability to explore the district level differences in community perspectives among men and women. However, the consistency of our results from across four districts show transferability across Bangladesh and other similar rural settings. Further research in urban settings may be helpful in consideration of unique access needs in those populations. During our IDIs with PE/E survivors, there were two instances where the survivor could not recall the details of her PE/E complications. Therefore, a

companion who was present when the survivor experienced those complications and was present during the IDI was able to communicate to researchers the details of the survivor's experience. Our study's selection of individual PE/E survivors interviewed precludes those who did not survive and pregnant women with other maternal complications, which limits our ability to make transferable claims about normative and individual perspectives broadly. However, we can infer that enabling internal and external influences to formal care access may be greater in our study sample than in the general population of women requiring care for maternal complications, reinforcing the need to address external influences through increased community awareness-building and primary healthcare strengthening efforts.

## Conclusion

This study shows variable pathways to care in rural Bangladesh and these pathways are highly influenced by internal and external factors that can be enabling or constraining. Our findings show that deaths from PE/E occur largely as a result of using one or more health systems and delays in seeking formal health services. The delays most common in our findings are a delay in deciding to seek care and arriving at a health facility. Receiving poor quality care at a facility was also reported, but was not as common in our sample. In addition to these delays, our findings indicate that accessing life-saving maternal health services is a complex process that may be complicated by the many players involved, such as non-medical staff at health facilities.

Pathways were compared across the four districts, and showed consistency in terms of contact points and type of facilities. These consistencies suggest the need to support quality improvement in primary healthcare and strengthen gender equity through community-based health promotion that raises awareness of the importance of birth preparedness and enhance women's autonomy and allows them to make decisions about their reproductive and maternal healthcare. These community-based health promotion activities can include targeted policy and programming that incorporates women's and men's health groups, religious leaders— trusted community members—and public health information in facilities that utilizes visual imagery of signs and symptoms of complications.

## Supporting information

**S1 Appendix.**
(TIF)

**S2 Appendix.**
(TIF)

## Author Contributions

**Conceptualization:** Pooja Sripad, Sharif Mohammed Ismail Hossain, Charlotte Warren.

**Data curation:** Kanij Sultana, Sharif Mohammed Ismail Hossain.

**Formal analysis:** Amy Dempsey, Pooja Sripad, Karen Kirk, Charlotte Warren.

**Project administration:** Kanij Sultana, Sharif Mohammed Ismail Hossain, Charlotte Warren.

**Supervision:** Kanij Sultana, Charlotte Warren.

**Visualization:** Karen Kirk.

**Writing – original draft:** Amy Dempsey, Pooja Sripad.

**Writing – review & editing:** Amy Dempsey, Pooja Sripad, Kanij Sultana, Karen Kirk, Sharif
   Mohammed Ismail Hossain, Charlotte Warren.

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
