## [Decision Letter · Decision Letter 0]

22 Apr 2020

PONE-D-19-24100

Pathways to service access for pre-eclampsia and eclampsia in rural Bangladesh: exploring women’s care-seeking

PLOS ONE

Dear Ms. Dempsey,

Thank you for submitting your manuscript to PLOS ONE. After careful consideration, we feel that it has merit but does not fully meet PLOS ONE’s publication criteria as it currently stands. Therefore, we invite you to submit a revised version of the manuscript that addresses the points raised during the review process.

You are encouraged to plan close attention to the comments from the reviewers. Please be aware a final decision will be made based on your response. We therefore as that you follow the PLOS ONE guideline for resubmission, making sure to specifically and cogently provide link answers in both a separate documents and in the text. 

We look forward to receiving your document.

We would appreciate receiving your revised manuscript by Jun 06 2020 11:59PM. To enhance the reproducibility of your results, we recommend that if applicable you deposit your laboratory protocols in protocols.io, where a protocol can be assigned its own identifier (DOI) such that it can be cited independently in the future. For instructions see: http://journals.plos.org/plosone/s/submission-guidelines#loc-laboratory-protocols

We look forward to receiving your revised manuscript.

Kind regards,

Joseph Telfair, DrPH, MSW, MPH

Academic Editor

PLOS ONE

Journal Requirements:

2. Please include a copy of the interview guide used in the study, in both the original language and English, as Supporting Information, or include a citation if it has been published previously.

Reviewers' comments:

Reviewer's Responses to Questions

**Comments to the Author**

1. Is the manuscript technically sound, and do the data support the conclusions?

Reviewer #1: Yes

Reviewer #2: Yes

2. Has the statistical analysis been performed appropriately and rigorously? 

Reviewer #1: N/A

Reviewer #2: N/A

3. Have the authors made all data underlying the findings in their manuscript fully available?

Reviewer #1: Yes

Reviewer #2: Yes

4. Is the manuscript presented in an intelligible fashion and written in standard English?

Reviewer #1: Yes

Reviewer #2: Yes

5. Review Comments to the Author

Reviewer #1: This is a report on qualitative study evaluating the lived experience of pre-eclampsia / eclampsia survivors in Bangladesh. The authors performed in-depth interviews with 22 affected women. They also conducted one male and one female focus group in each district (4 districts, i.e. 8 focus groups) with community members. The main finding is that women have variable pathways of care and that these are highly influenced by internal and external factors. Based on the findings, the authors are appropriately calling for quality improvement in primary health care focused on better birth-preparedness and maternal health. Some minor suggestions for improvements are listed here:

Abstract:

The abstract is overall well written. However, the conclusion is hard to understand. Would suggest rewording the conclusion.

Introduction:

The introduction gives a good background on the topic and the rationale to perform the study.

Methods:

1. A sample size justification would be helpful. Why were 22 women selected for interviews?

2. Data collection: a bit more detail on how the focus groups and interviews were conducted would be helpful. Ideally, the authors could provide the interview / focus group guides in an appendix (in local language and English translation).

3. Data analyses: who did the coding and what was the coders background / expertise related to qualitative analyses and pre-eclampsia?

4. Data analyses: were any measures taken to assure rigor and validity in the qualitative analyses? For example, double coding, triangulation, team discussions, member checking, searching for disconfirming evidence?

Results:

1. The legend for Figure 3 is very helpful, but this is only attached to Figure 3 right now. Would suggest adding it to Figures 1 and 2 or combining all three figures into one figure.

2. How were the 7 women shown in Figures 1, 2, and 3 selected out of the total of 22?

3. When reading the results, I was surprised to see in-depth interviews with others than the PE/E survivors. For example, there are quotes from a survivor’s neighbor and from a survivor’s mother. I would suggest adding to the methods how these participants were sampled and how many of them were interviewed.

Discussion:

1. The discussion appropriately summarizes the results and puts them into perspective with the current literature.

2. The conclusions are appropriate and largely supported by the data. The only part where not much data is shown is around the consistency of findings across districts. Would suggest adding a brief section on how consistency across districts was evaluated to the methods and also some sentences stating these findings to the results.

Reviewer #2: This is a strong manuscript with a great methodology. The schematic representation of pathways to care are very insightful.

I am going to comment on the methods, which are strong.

1 - You did not refer to or cite grounded theory in the body of the paper, only in the abstract. It is evident that you used a strong approach to thematic analysis. Please delete reference to grounded theory in the abstract. I would recommend wording that indicates (a) the total number of focus groups, (b) representation of male and female perspectives, (c) analysis of pathways through the health system, (d) careful thematic analysis using Atlas-ti.

Some slight improvements:

1 - Method - Please indicate types of question format (open ended, semi-structured, etc.)

2 - Add a citation for the thematic analysis.

3 - reference to purposive selection - how were initial participants found? Advertising? Clinical records?

2 - Barriers to care:

The paper would be enriched by some reference to how the participants travelled. is often absent in papers on rurality, which makes it hard to compare one "rural" paper to another "rural" paper. Bangladesh is a very densely populated rural country, matching central China and areas in Southeast Asia, but not similar to Scotland and rural Australia or mountain regions of South America, for example. Indicate density (e.g. avg. population per sq km for the counties you conducted your research in)

A - How do they travel ? how long? multi-modal? cars? on foot? public transport?

B - Did women have control of these modes of transport? This is critical in understanding pathways to care.

C - Was finance a barrier to travel?

D - how did they travel during crisis? For example, this is a powerful quote, but leaves an information gap in understanding how she actually travelled:

"“Her eyes were rolling and she was biting her lips! And the seizures started. We were told

to take her to Dhaka Medical College, otherwise she would not survive because [the

facility] didn’t have any medicines.” – Female, IDI

For the importance of transport, see this (I have no connection):

Brown, H., Varcoe, C., & Calam, B. (2011). The birthing experience of rural aboriginal women in context: Implications for nursing. Canadian Journal of Nursing Research, 43, 100-117.

3 - Recommendations / Conclusions

Your recommendations are too general. What I see as important issues, based on your findings:

A - as you point out, lack of prior awareness, and therefore the need to be more vigorous in education

B - your findings indicate intergenerational beliefs, particularly among the older generation of women who reference faith-based explanations to dismiss need for antenatal care. Would it make sense to work with faith leaders to help to begin to change beliefs? This is a powerful source of family stability and support, so women will not want to cause trouble, yet they might be in danger. What is an appropriate approach to addressing this dilemma?

C - it seems that systemic issues at the hospitals, lack of referrals, and unavailability of physicians are major barriers. Are there systemic ways to address this beyond recruiting more physicians? e.g. community doulas who can help navigate systems?

The methodology was very well explained, and the quotes and presentation of information is clear and powerful. Thank you.

6. PLOS authors have the option to publish the peer review history of their article (what does this mean?). If published, this will include your full peer review and any attached files.

Reviewer #1: Yes: Florian R. Schroeck, MD, MS

Reviewer #2: Yes: Silvia L Vilches

---

## [Author Response · Author response to Decision Letter 0]

15 Jul 2020

Dr. Joerg Hever

Editor in Chief, PLOS ONE

1160 Battery Street 

Koshland Building East, Suite 225

San Francisco, CA 94111

May 24, 2020

Dear Dr. Hever, 

Thank you for the review of the manuscript entitled, “Pathways to service access for pre-eclampsia and eclampsia in rural Bangladesh: exploring women’s care-seeking.” We hope this revised manuscript is acceptable for publication in PLOS ONE. 

We are grateful for the reviewers’ comments and are glad they feel the presentation of pre-eclampsia and eclampsia survivors’ experiences shared in this manuscript is clear and powerful. The focus on these women’s personal stories of accessing maternal health services while experiencing pregnancy-related complications is imperative for improving community perceptions of formal health systems and increasing access to lifesaving care. We welcome the chance to continue improving our manuscript. Please find below detailed responses to the reviewer’s comments and descriptions of corresponding revisions made in the resubmitted manuscript. 

This manuscript comes from original work and is not published or under consideration for publication elsewhere. The authors have no competing interests, and we have approved the manuscript for submission.

We thank you for your consideration and look forward to hearing from you.

Sincerely,

Amy Dempsey, MA

Population Council

4301 Connecticut Ave NW, Suite 280

Washington, DC 20008

Reviewer #1: This is a report on qualitative study evaluating the lived experience of pre-eclampsia / eclampsia survivors in Bangladesh. The authors performed in-depth interviews with 22 affected women. They also conducted one male and one female focus group in each district (4 districts, i.e. 8 focus groups) with community members. The main finding is that women have variable pathways of care and that these are highly influenced by internal and external factors. Based on the findings, the authors are appropriately calling for quality improvement in primary health care focused on better birth-preparedness and maternal health. Some minor suggestions for improvements are listed here: 

Comment: The abstract is overall well written. However, the conclusion is hard to understand. Would suggest rewording the conclusion.

Response: Thank you for this feedback. Our proposed, edited language is below.

Page 2: Women’s individual pathways to care were diverse, despite the homogenous community perceptions of the influences on women’s care-seeking behaviors. This finding supports the need for improving quality of care in primary health care facilities and strengthening gender equity and community-based promotion activities through targeted policy and programming. 

Comment: In the Methods, a sample size justification would be helpful. Why were 22 women selected for interviews?

Response: We have updated the language to provide more detail on how these 22 women selected. 

Page 5:Participants were a convenient sampling selected based on their lived experiences of maternal complications and diversity in care-seeking pathways in rural Bangladesh. All participants were recruited from tertiary hospitals, where they received care after initiating PE/E services in different regions of Bangladesh. 

Comment: Data collection: A bit more detail on how the focus groups and interviews were conducted would be helpful. Ideally, the authors could provide the interview / focus group guides in an appendix (in local language and English translation).

Response: We have added the Bangla (local language) and English versions of the IDI and FGDs guides as appendices. We also updated some language on page 6 that points the reader to the appendices for these guides. 

Pg. 6: 

Before beginning the study, approval and administrative orders were collected from Directorate General of Family Planning (DGFP) and Directorate General of Health Services (DGHS). Hospital ethical approval was obtained from the hospital ethical committee. The data collection team was recruited and trained on the study objectives, tools, procedures, and research ethics in dealing with human subjects. When possible, the data collectors were women with prior experience interviewing women on sensitive issues. Participants received information on informed consent procedures before the data collectors conducted interviews. The IDI and FGD guides are attached as appendices. 

Comment: Data analyses: who did the coding and what was the coders background / expertise related to qualitative analyses and pre-eclampsia?

Response: We have added clarifying language in response to this language. 

Pg. 6: Researchers analyzed data according to rigorous methods and used a thematic analysis approach that drew on emergent themes and the research questions (13). Interviews were digitally audio recorded and supported by hand-written notes. Transcripts were transcribed verbatim and translated into English for content. An inductively derived codebook was developed after reading the IDI and FGD transcripts and generating themes based on those transcripts. One team member applied a codebook using Atlas-ti. The team discussed the outputs of these codes, which were the catalyst for this manuscript about the pathways to care from individual and group-normative perspectives.

Coders had a social science or public health background, were intimately involved in the Ending Eclampsia project, and were very familiar with the concepts of PE/E research. Influences on care seeking pathways emerged as areas of interest. In this paper, we describe two types of influences: 1) Internal influences defined as a woman making a care-seeking decision based on her own knowledge and ability, and without consultation or advice from her family, neighbors or community; and 2) External influences defined as care-seeking decisions made based on consultation and/or advice from the family, neighbors or community, as well as perceptions of whether the health system can provide quality care. 

Comment: Data analyses: were any measures taken to assure rigor and validity in the qualitative analyses? For example, double coding, triangulation, team discussions, member checking, searching for disconfirming evidence?

Response: We have added clarifying language in response to this. 

Pg. 6: Interviews were digitally audio recorded and supported by hand-written notes. Transcripts were transcribed verbatim and translated into English for content. An inductively derived codebook was developed after reading the IDI and FGD transcripts and generating themes based on those transcripts. One team member applied a codebook using Atlas-ti. The team discussed the outputs of these codes, which were the catalyst for this manuscript about the pathways to care from individual and group-normative perspectives. 

Coders had a social science or public health background, were intimately involved in the Ending Eclampsia project, and were very familiar with the concepts of PE/E research. Influences on care seeking pathways emerged as road domains of interest. In this paper, we describe two types of influences: 1) Internal influences defined as a woman making a care-seeking decision based on her own knowledge and ability and without consultation or advice from her family, neighbors or community; and 2) External influences defined as care-seeking decisions made based on consultation and/or advice from the family, neighbors or community, as well as perceptions of whether the health system can provide quality care. 

Preliminary outputs and themes were confirmed and refined during in-country dissemination meetings with critical maternal health stakeholders in the Bangladeshi maternal health, specifically PE/E, sector. This served as an initial member-checking step to ensure credibility of the data.

Comment: Results: The legend for Figure 3 is very helpful, but this is only attached to Figure 3 right now. Would suggest adding it to Figures 1 and 2 or combining all three figures into one figure.

Response: Each of these figures have a legend, and have been re-uploaded as separate documents. 

Comment: How were the 7 women shown in Figures 1, 2, and 3 selected out of the total of 22?

Response: We have added language that addresses this comment and explains that the examples shown in these figures represent a full range of women’s experiences. 

Pg. 7: These are examples that show the full range of women’s care-seeking experiences. These pathways varied and showed there is not one typical pathway to care. Rather, they involved combinations of staying home without seeking care, seeking advice from family and friends, visiting traditional healers, and making contact with the health system at different points in their pregnancy. 

Comment: When reading the results, I was surprised to see in-depth interviews with others than the PE/E survivors. For example, there are quotes from a survivor’s neighbor and from a survivor’s mother. I would suggest adding to the methods how these participants were sampled and how many of them were interviewed.

Response: In those examples, the PE/E survivors were unable to recall the details of their experiences. The companions who supported them during the IDIs were also present when the survivors experienced their complications. We have added language that explains that context in the Methods and Discussion sections. 

In Methods, Pg. 6: During the IDIs, there were two examples where the survivor could not recall the details of her PE/E complications given the unconscious state she was in during that time. A companion who was present when the survivor experienced those complications and was present during the IDI communicated to researchers the missing details of the survivor’s pathway and experience

In Discussion, Pg. 14: Limitations of this study include our inability to explore the district level differences in community perspectives among men and women. However, the consistency of our results from across four districts show transferability across Bangladesh and other similar rural settings. Further research in urban settings may be helpful in consideration of unique access needs in those populations. During our IDIs with PE/E survivors, there were two instances where the survivor could not recall the details of her PE/E complications. Therefore, a companion who was present when the survivor experienced those complications and was present during the IDI was able to communicate to researchers the details of the survivor’s experience. Our study’s selection of individual pre-eclampsia survivors interviewed precludes those who did not survive and pregnant women with other maternal complications, which limits our ability to make transferable claims about normative and individual perspectives broadly. However, we can infer that enabling internal and external influences to formal care access may be greater in our study sample than in the general population of women requiring care for maternal complications, reinforcing the need to address external influences through increased community awareness-building and primary health care strengthening efforts.

Comments: 1. The discussion appropriately summarizes the results and puts them into perspective with the current literature.

2. The conclusions are appropriate and largely supported by the data. The only part where not much data is shown is around the consistency of findings across districts. Would suggest adding a brief section on how consistency across districts was evaluated to the methods and some sentences stating these findings to the results. 

Response: These pathways were compared across districts and were consistent in terms of contact points and type of facility visited by survivors. We have added language to the Conclusion and text that explains figures 1, 2, and 3 to clarify this. 

Pg. 8, Figure 1: Figure 1 shows one woman (A), who did not seek any ANC, but visited a traditional healer before when she experienced complications during delivery. As she experienced complications, she then traveled to a primary facility, where she received care from a SACMO, then to another PHC from an FWA and then back to the SACMO. She then went to a tertiary hospital for delivery, and after being discharged from the hospital, she did not seek any additional care in the postnatal period. Another woman (B) initially chose to forego ANC at a primary healthcare facility and instead sought advice from family and friends, but later visited a SACMO at a primary facility for ANC. The same woman experienced complications during delivery, and again sought advice from family and friends, then visited a SACMO at a primary facility, and from there went between the tertiary facility and a private hospital until she had a total of 11 points of contact with the health system. Neither of them attended PNC. There was no difference in the ways in which these pathways arose in the four districts. 

Pg. 9: Figure 2: Figure 2 shows women C, D, and E. Woman C had five ANC contacts with the tertiary hospital, where she also delivered her baby. Woman D started her pathway to PE/E care when she visited a tertiary hospital after experiencing complications during delivery. Woman E attended ANC at a private hospital first, and then went to a SACMO at primary facility once and to an FWV at primary facility twice. During delivery, when she experienced complications, she traveled back to the private hospital where she attended ANC, then back to a SACMO at a primary facility then to a tertiary facility. None of these women attended any PNC. These pathways were seen across the four districts. 

Pg. 9, Figure 3: In Figure 3, Woman F first sought advice from friends and family, then visited a traditional healer twice, without receiving any ANC. She then went from a tertiary hospital, to an FWV at a primary facility, then to a private hospital, back to the primary facility and then to a tertiary hospital. Woman G first sought advice from family and friends, then went for two ANC visits at a primary health center and then experienced complications related to PE/E during delivery and went to a tertiary facility and then to a primary facility, where she delivered her baby. After delivery, she again experienced complications and bounced between private, primary, and tertiary facilities for postnatal complications for eight points of contact with health facilities. These pathways were the same across the four districts.

Pg. 15: This study shows variable pathways to care in rural Bangladesh that are highly influenced by internal and external factors that can be enabling or constraining. Pathways were compared across the four districts, and showed consistency in terms of contact points and type of facilities. These consistencies suggest the need to support quality improvement in primary health care and strengthen gender-equity and community-based health promotion around better birth-preparedness of families and enhancing women’s autonomy within their household to make reproductive and maternal health care decisions through targeted policy and programming.

Reviewer #2: This is a strong manuscript with a great methodology. The schematic representation of pathways to care are very insightful. I am going to comment on the methods, which are strong.

Comment: You did not refer to or cite grounded theory in the body of the paper, only in the abstract. It is evident that you used a strong approach to thematic analysis. Please delete reference to grounded theory in the abstract. I would recommend wording that indicates (a) the total number of focus groups, (b) representation of male and female perspectives, (c) analysis of pathways through the health system (d) careful thematic analysis using Atlas-ti. 

Response: We have edited the language (removed grounded theory) in the abstract to reflect this comment. 

(Abstract) We conducted 22 in-depth interviews (IDIs) with pre-eclampsia and eclampsia survivors in tertiary hospitals, where they sought care from different parts of the country. In each of the four districts, we conducted one male and one female focus group discussion (FGD) to unearth care-seeking pathways and explore through individual and normative perspectives, the range of internal and external influences in four districts in Bangladesh. Careful thematic analysis using Atlas-ti was applied.

Comment: Methods - Please indicate types of question format (open ended, semi-structured, etc.)

Response: These questions were open-ended and exploratory. Researchers asking probing questions for more specificity and detailed responses. 

Pg. 4: The study adopted a cross sectional qualitative approach that is best suited to exploring in-depth perspectives of individual narratives to unearth care-seeking pathways and their complexity and community views on pathways to accessing care in the Bangladeshi context. We conducted focus group discussions (FGDs) and in-depth interviews (IDIs) of PE/E survivors to better understand user perceptions around PE/E, including their knowledge about the condition as well as the barriers and enablers of accessing antenatal and delivery care. The IDIs and FGDs consisted of open-ended and exploratory questions, with researchers asking probing questions for more specific and detailed responses. 

Comment: Add a citation for the thematic analysis. 

Response: We have added more language and a citation for thematic analysis. 

Researchers analyzed data according to rigorous methods and used a thematic analysis approach that drew on emergent and research questions (13). Interviews were digitally audio recorded and supported by hand-written notes. Transcripts were transcribed verbatim and translated into English for content. An inductively derived codebook was developed after reading the IDI and FGD transcripts and generating themes based on those transcripts. One team member applied a codebook using Atlas-ti. The team discussed the outputs of these codes, which were the catalyst for this manuscript about the pathways to care from individual and group-normative perspectives. 

Comment: Reference to purposive selection - how were initial participants found? Advertising? Clinical records? 

Response: We have updated to language to provide more detail on how these 22 women selected. 

Pg. 5: Participants were a convenient sampling selected based on their lived experiences of maternal complications and diversity in care-seeking pathways in rural Bangladesh. They were recruited from a tertiary hospital, where they received care for PE/E after initiating services in different regions of Bangladesh. To elicit community perspectives on knowledge, beliefs, and behaviors related to pregnancy and childbirth, researchers conducted eight FGDs (7-9 persons per group) with married men (age 18-55 years) and with women (18-49 years) who had at least one living child. One female and one male FGD were conducted in each district. 

Comment: Barriers to care: The paper would be enriched by some reference to how the participants travelled. is often absent in papers on rurality, which makes it hard to compare one "rural" paper to another "rural" paper. Bangladesh is a very densely populated rural country, matching central China and areas in Southeast Asia, but not similar to Scotland and rural Australia or mountain regions of South America, for example. Indicate density (e.g. avg. population per sq km for the counties you conducted your research in) 

Response: We have added information on population density per square kilometer of the districts we conducted research. 

Pg. 4: Bangladesh is a homogenous country in terms of culture, language, cultural norms, education, health facility and ethnicity. The study was conducted in four districts: Tangail, which has a population density of 975 people per square kilometer; Bhola, which has a population density of 456 people per square kilometer; Rangpur, which has a population density of 1,101 people per square kilometer; and Cumilla, which has a population density of 1,290 people per square kilometer. These districts were chosen after consultation with the Ministry of Health. Cumilla is southeast of Dhaka and consists of 16 upazilas, with most people engaged in agriculture, trade and textile work. Tangail consists of 12 upazilas. With close proximity to Dhaka, the urban growth rate in these two districts is increasing. With eight upazilas, Rangpur is in northern Bangladesh and Bhola is an island located in the south-central part of the country and has seven upazilas. 

Comment: How do they travel? how long? multi-modal? cars? on foot? public transport?

Response: We agree that understanding the ways in which women in rural settings travel to health facilities is very important. We have added new language to explain how the women in figures 1, 2, and 3 traveled. 

Pg. 8: Figure 1 shows one woman (A), who did not seek any ANC, but visited a traditional healer before when she experienced complications during delivery. As she experienced complications, she then traveled to a primary facility by car, where she received care from a SACMO, then to another PHC from an FWA and then back to the SACMO. She then went to a tertiary hospital by car for delivery, and after being discharged from the hospital, she did not seek any additional care in the postnatal period. Another woman (B) initially chose to forego ANC at a primary healthcare facility and instead sought advice from family and friends, but later traveled by car to a SACMO at a primary facility for ANC. The same woman experienced complications during delivery, and again sought advice from family and friends, then visited a SACMO at a primary facility, and from there went between the tertiary facility and a private hospital until she had a total of 11 points of contact with the health system. She delivered her baby at the tertiary hospital, and traveled there by ambulance. Neither of these women attended PNC. There was no difference in ways in which these pathways arose in the four districts.

Pg. 9: Figure 2 shows women C, D, and E. Woman C had five ANC contacts with the tertiary hospital, where she also delivered her baby. Woman D started her pathway to PE/E care by traveling to a tertiary hospital after experiencing complications during delivery. Woman E attended ANC at a private hospital first, then went to a SACMO at primary facility once, and to an FWV at primary facility twice. During delivery, when she experienced complications, she traveled back to the private hospital where she attended ANC, then back to a SACMO at a primary facility then to a tertiary facility. None of these women attended any PNC, and they traveled to these health facilities by car or by bus. These pathways were seen across the four districts.

Pg. 9: In Figure 3, Woman F first sought advice from friends and family, then visited a traditional healer twice, without receiving any ANC. She then traveled in a rickshaw a tertiary hospital, to an FWV at a primary facility, then to a private hospital. She then went back to the primary facility, and from there was taken to a tertiary hospital. Woman G first sought advice from family and friends, then went for two ANC visits at a primary health center and then experienced complications related to PE/E during delivery and went to a tertiary facility and then to a primary facility, where she delivered her baby. After delivery, she again experienced complications and bounced between private, primary, and tertiary facilities for postnatal complications for eight points of contact with health facilities. These pathways were the same across the four districts.

Comment: Did women have control of these modes of transport? This is critical in understanding pathways to care. 

Response: We have added language on page 11 that clarifies who has decision-making power during pregnancy and when pregnancy-related complications arise. Typically, husbands, mothers-in-law, and religious leaders are the decision-makers. 

Pg. 11: In many cases, women are not empowered to make their own decisions about seeking and receiving health services, even during pregnancy. The FGDs showed that these decisions are often made by husbands, who as they are often the sole financial contributor to a family, and thus the sole decision maker in a family. The FGDs also describe religious leaders and mothers-in-law as trusted, respected members of their communities, making them key influencers of women’s health-seeking behaviors during pregnancy. 

Comment: Was finance a barrier to travel? 

Response: Finances are often a barrier to traveling to health facilities, and have added language to pg. 11 that clarifies this point. 

Pg. 11: The FGDs showed that husbands often make these decisions, as they are often the sole financial contributor to a family, and thus the sole decision maker in a family.

Comment: How did they travel during crisis? For example, this is a powerful quote, but leaves an information gap in understanding how she actually traveled: "“Her eyes were rolling and she was biting her lips! And the seizures started. We were told to take her to Dhaka Medical College, otherwise she would not survive because [the facility] didn’t have any medicines.” – Female, IDI

For the importance of transport, see this (I have no connection): Brown, H., Varcoe, C., & Calam, B. (2011). The birthing experience of rural aboriginal women in context: Implications for nursing. Canadian Journal of Nursing Research, 43, 100-117.

Response: We agree that transport is extremely important. We have clarified this in the Discussion section on pg. 13. 

Pg. 13: Prevailing views of women and communities in rural Bangladesh indicate varied pathways to care throughout their pregnancy, during childbirth, and in the postnatal period influenced by internal and external factors at the individual, familial, social, and health systems levels. Internal factors, such as women’s awareness of relevant complications and options with respect to hypertension and PE/E in particular, as well as their financial means and autonomous ability to make a decision to seek care, were often modified by external factors. Socio-cultural norms around restricted female mobility, childbirth-oriented practices of hiding one’s pregnancy, and the role of spouses, mothers-in-law, neighbors somewhat hindered women’s access to ANC, but often were overcome at the delivery stage, when a woman experienced more visible complications. Respondents lived in rural settings, and similar to other research (18) that examined the impact rurality has on maternal health outcomes, this influenced their pathways to care and the ways in which they traveled to health facilities, which included diverse transportation methods that ranged from cars to rickshaws to buses. Health systems deficiencies, including provider absenteeism at primary health care facilities 24 hours compound these delays and often lead to variable pathways to care (9).

Comment: Your recommendations are too general. What I see as important issues, based on your findings…as you point out, lack of prior awareness, and therefore the need to be more vigorous in education.

Comment: Your findings indicate intergenerational beliefs, particularly among the older generation of women who reference faith-based explanations to dismiss need for antenatal care. Would it make sense to work with faith leaders to help to begin to change beliefs? This is a powerful source of family stability and support, so women will not want to cause trouble, yet they might be in danger. What is an appropriate approach to addressing this dilemma?

Response: Thank you for this feedback. We have added some specificity around the importance of education and adjusted the language here related to intergenerational perceptions to bolster the recommendations. 

Pgs. 14 – 15: Key policy/program recommendations that emerge out of these findings are similar to research findings from Nigeria (16) and center around the importance of brining health services closer to the people who need them. Enabling earlier ANC use and empowering women and their families to identify signs and symptoms of PE/E/, among other maternal complications, and seek care will to minimize the risk of adverse maternal and newborn outcomes. This could be implemented through increased and targeted educational strategies at the community level. Non-distinct patterning of survivor perspectives across Tangail, Bhola, Rangpur, and Cumilla districts suggests that similar strategic programming may be applied and scaled in relatively homogenous populations in rural Bangladesh. Moreover, the larger gap in community knowledge merits attention in terms of program targets and implies the need to strengthen the role of community health systems in maternal health.

There are intergenerational differences between younger and older women and their perceptions of the formal health system and beliefs about seeking formal health services, particularly during pregnancy and childbirth. This significantly influences how families support women’s care seeking deserves further investigation. Additionally, community awareness building around hypertension in pregnancy, dispelling myths around childbirth complications, working with faith leaders to change perceptions of the formal health system, emphasizing among male-partners and joint-family members more equitable health care-seeking decisions in the home are paramount. 

Comment: It seems that systemic issues at the hospitals, lack of referrals, and unavailability of physicians are major barriers. Are there systemic ways to address this beyond recruiting more physicians? e.g. community doulas who can help navigate systems?

Response: We have added additional language that discusses the importance of bringing health services closer to the people who need them, and empowering women and their families to access early ANC and recognize signs and symptoms of pregnancy-related complications. 

Pg. 14: Key policy/program recommendations that emerge out of these findings are similar to research findings from Nigeria (16) and health services closer to the people who need them. Enabling earlier ANC use and empowering women and their families to identify signs and symptoms of PE/E/, among other maternal complications, and seek care will minimize the risk of adverse maternal and newborn outcomes. Non-distinct patterning of survivor perspectives across Tangail, Bhola, Rangpur, and Cumilla districts suggests that similar strategic programming may be applied and scaled in relatively homogenous populations in rural Bangladesh. Moreover, the larger gap in knowledge at the community level merits attention in terms of program targets. 

Comment: The methodology was very well explained, and the quotes and presentation of information is clear and powerful. Thank you.

---

## [Decision Letter · Decision Letter 1]

25 Aug 2020

PONE-D-19-24100R1

Pathways to service access for pre-eclampsia and eclampsia in rural Bangladesh: exploring women’s care-seeking

PLOS ONE

Dear Dr. Dempsey,

Thank you for submitting your manuscript to PLOS ONE. After careful consideration, we feel that it has merit but does not fully meet PLOS ONE’s publication criteria as it currently stands. Therefore, we invite you to submit a revised version of the manuscript that addresses the points raised during the review process.

The authors have shown significant improvement of the manuscript. The author needs to address the comments from the second reviewer - if satisfactory, it can be deem adequate for potential publication.

We look forward to receiving your revised manuscript.

Kind regards,

Joseph Telfair, DrPH, MSW, MPH

Academic Editor

PLOS ONE

Reviewers' comments:

Reviewer's Responses to Questions

**Comments to the Author**

1. If the authors have adequately addressed your comments raised in a previous round of review and you feel that this manuscript is now acceptable for publication, you may indicate that here to bypass the “Comments to the Author” section, enter your conflict of interest statement in the “Confidential to Editor” section, and submit your "Accept" recommendation.

Reviewer #1: All comments have been addressed

Reviewer #2: All comments have been addressed

2. Is the manuscript technically sound, and do the data support the conclusions?

Reviewer #1: Yes

Reviewer #2: Yes

3. Has the statistical analysis been performed appropriately and rigorously? 

Reviewer #1: N/A

Reviewer #2: N/A

4. Have the authors made all data underlying the findings in their manuscript fully available?

Reviewer #1: Yes

Reviewer #2: Yes

5. Is the manuscript presented in an intelligible fashion and written in standard English?

Reviewer #1: Yes

Reviewer #2: Yes

6. Review Comments to the Author

Reviewer #1: (No Response)

Reviewer #2: The authors have taken care to respond to reviewers' comments and the comments are well-addressed. The manuscript is much clearer, especially the methodology. The presentation of method is very consistent with a strong thematic analysis. Wellcdone.

I have minor comments, which could strengthen the manuscript.

***

1. The correct term is “convenience sample” not “convenient sampling” (even though this makes more grammatical sense).

**

2. The following sentence is awkward (pp. 14-15):

"Our findings align with other studies that examined maternal healthcare seeking among adolescent girls in Bangladesh, suggesting that women’s lack of knowledge of signs and symptoms and the

15importance of ANC may have a larger interactive effect with internal processing of decisions among younger women (14), as well as socio-economic status (15).

Suggestion:

"Our findings align with other studies that examined maternal healthcare seeking behavior among adolescent girls in Bangladesh, suggesting that women’s lack of knowledge about the importance of ANC and its of signs and symptoms may negatively? interact with socio-economic status of women (15) and affect internal processing of decisions among younger women (14).

• Note: You do not indicate what the effect is of the interaction with socioeconomic status, so I’m guessing that it compounds disadvantage. Please correct as needed.

**

3. This sentence would be better placed in the methodology, rather than the conclusion. It is a clarification, not a limitation.

“During our IDIs with PE/E survivors, there were two instances where the survivor could not recall the details of her PE/E complications. Therefore, a companion who was present when the survivor experienced those complications and was present during the IDI was able to communicate to researchers the details of the survivor’s experience.

**

4. (a) There is mention of gendered roles in the findings, but no real discussion, until the conclusion. What, specifically, could be done to assist men to make better decisions?

4. (b) The same occurs with lower levels of education: you present it, but do not discuss it. However, you mentions that some women were aware of E/PE through brochures; could public health information help, even for less-educated women?

**

5. On p. 3, you outlines the following…

“In addition to clinical and health systems issues that negatively impact pregnant women and their families, deaths from PE/E occur largely as a result of delays in seeking health services (4)andthe use of one or more health systems (5,6). Thaddeus and Maine posit that delays in seeking care can occur at three different points: (1) delay in deciding to seek care, (2) delay in arriving at a health facility, and (3) delay in receiving quality care at a facility (7). This paper focuses primarily on the first and second delays, a…

Suggestion: In the conclusion, can you reflect back and connect your results to these propositions? It would make your paper stronger if you contextualized your findings with others’ prior studies. You have some evidence, as well, that proposition three is confirmed; that there is delay in receiving services at some facilities, either due to inconsistence service or pay barriers. This seems to be complicated by the involvement of nonmedical staff as gatekeepers.

**

6. In the introduction, you state that:

“…[in] Bangladesh, between 5,000 and 6,000 women die during pregnancy or childbirth, and up to 1,200 –or 24percent –of those deaths are from eclampsia (10), the second most common cause of maternal mortality after postpartum hemorrhage.”

Suggestion: Can you reflect back, in your conclusion, on how much of a difference it might make to tackle various problems? For example, is the primary barrier to better treatment the inability of facilities to provide care? Or is the primary problem lack of knowledge among men (husbands and fathers) and women (mothers, mothers-in-law, etc.) about E/PE; or is the primary problem the circling back and forth between different types of providers (traditional healers, local health clinics, etc)? Your reflections might not be strictly evidence, but you know your material better than anyone else – what are your thoughts?

**

7. Question: How different is this situation than it is in urban Bangladesh? Are these problems of lack of access and knowledge purely rural? I suspect not, so in that case, how can these findings help those in urban areas? Should this be investigated further? Could you make a recommendation for further research?

Your findings will be of interest to rural studies. Please consider using "rural" as a keyword

7. PLOS authors have the option to publish the peer review history of their article (what does this mean?). If published, this will include your full peer review and any attached files.

Reviewer #1: No

Reviewer #2: **Yes: **Silvia L. Vilches

---

## [Author Response · Author response to Decision Letter 1]

3 Oct 2020

Dr. Joerg Hever

Editor in Chief, PLOS ONE

1160 Battery Street 

Koshland Building East, Suite 225

San Francisco, CA 94111

October 1, 2020

Dear Dr. Hever, 

Thank you for the review of the manuscript entitled, “Pathways to service access for pre-eclampsia and eclampsia in rural Bangladesh: exploring women’s care-seeking.” We hope this revised manuscript is acceptable for publication in PLOS ONE. 

We are grateful for the reviewers’ comments and are glad they feel the presentation of pre-eclampsia and eclampsia survivors’ experiences shared in this manuscript is clear and powerful. The focus on these women’s personal stories of accessing maternal health services while experiencing pregnancy-related complications is imperative for improving community perceptions of formal health systems and increasing access to lifesaving care. We welcome the chance to continue improving our manuscript. Please find below detailed responses to the reviewer’s comments and descriptions of corresponding revisions made in the resubmitted manuscript. 

This manuscript comes from original work and is not published or under consideration for publication elsewhere. The authors have no competing interests, and we have approved the manuscript for submission

We thank you for your consideration and look forward to hearing from you.

Sincerely,

Amy Dempsey, MA

Population Council

4301 Connecticut Ave NW, Suite 280

Washington, DC 20008

Comment: The correct term is “convenience sample” not “convenient sampling” (even though this makes more grammatical sense).

Response: We have changed this to “were from a convenience sample”

Comment: The following sentence is awkward (pp. 14-15): "Our findings align with other studies that examined maternal healthcare seeking among adolescent girls in Bangladesh, suggesting that women’s lack of knowledge of signs and symptoms and the (15) importance of ANC may have a larger interactive effect with internal processing of decisions among younger women (14), as well as socio-economic status (15).Note: You do not indicate what the effect is of the interaction with socioeconomic status, so I’m guessing that it compounds disadvantage. Please correct as needed.

Response: We have changed this to say “Our findings align with other studies that examined maternal healthcare seeking behavior among adolescent girls in Bangladesh, suggesting that women’s lack of knowledge about the importance of ANC and its signs and symptoms may negatively interact with socio-economic status of women (15) and affect internal processing of decisions among younger women (14).”

Comment: This sentence would be better placed in the methodology, rather than the conclusion. It is a clarification, not a limitation. “During our IDIs with PE/E survivors, there were two instances where the survivor could not recall the details of her PE/E complications. Therefore, a companion who was present when the survivor experienced those complications and was present during the IDI was able to communicate to researchers the details of the survivor’s experience.

Response: Thank you for this comment. This sentence was in both places already - methodology (page 6) and limitations (page 16). 

Comments 4 & 5: 

4. (a) There is mention of gendered roles in the findings, but no real discussion, until the conclusion. What, specifically, could be done to assist men to make better decisions?

4. (b) The same occurs with lower levels of education: you present it, but do not discuss it. However, you mentions that some women were aware of E/PE through brochures; could public health information help, even for less-educated women?

5. On p. 3, you outlines the following…

“In addition to clinical and health systems issues that negatively impact pregnant women and their families, deaths from PE/E occur largely as a result of delays in seeking health services (4)andthe use of one or more health systems (5,6). Thaddeus and Maine posit that delays in seeking care can occur at three different points: (1) delay in deciding to seek care, (2) delay in arriving at a health facility, and (3) delay in receiving quality care at a facility (7). This paper focuses primarily on the first and second delays, a…

Suggestion: In the conclusion, can you reflect back and connect your results to these propositions? It would make your paper stronger if you contextualized your findings with others’ prior studies. You have some evidence, as well, that proposition three is confirmed; that there is delay in receiving services at some facilities, either due to inconsistent service or pay barriers. This seems to be complicated by the involvement of nonmedical staff as gatekeepers. 

Response to comments 4 - 5: We have updated this section to say the following: 

This study shows variable pathways to care in rural Bangladesh and these pathways are highly influenced by internal and external factors that can be enabling or constraining. Our findings show that deaths from PE/E occur largely as a result of using one or more health systems and delays in seeking health services. The delays most common in our findings are a delay in deciding to seek care and arriving at a health facility. Receiving poor quality care at a facility was also reported, but was not as common in our sample. In addition to these delays, our findings indicate that accessing life-saving maternal health services is a complex process that may be complicated by the many players involved, such as non-medical staff at health facilities. 

Pathways were compared across the four districts, and showed consistency in terms of contact points and type of facilities. These consistencies suggest the need to support quality improvement in primary healthcare and strengthen gender equity through community-based health promotion that raises awareness of the importance of birth preparedness and enhance women’s autonomy and allows them to make decisions about their reproductive and maternal health care. These community-based health promotion activities can include targeted policy and programming that incorporates women’s and men’s health groups, religious leaders - trusted community members - and public health information in facilities that utilizes visual imagery of signs and symptoms of complications.

Comment: In the introduction, you state that: “…[in] Bangladesh, between 5,000 and 6,000 women die during pregnancy or childbirth, and up to 1,200 –or 24percent –of those deaths are from eclampsia (10), the second most common cause of maternal mortality after postpartum hemorrhage.”

Suggestion: Can you reflect back, in your conclusion, on how much of a difference it might make to tackle various problems? For example, is the primary barrier to better treatment the inability of facilities to provide care? Or is the primary problem lack of knowledge among men (husbands and fathers) and women (mothers, mothers-in-law, etc.) about E/PE; or is the primary problem the circling back and forth between different types of providers (traditional healers, local health clinics, etc)? Your reflections might not be strictly evidence, but you know your material better than anyone else – what are your thoughts?

Response: We have updated this section to say: There is no single barrier to accessing maternal health services, but rather multiple barriers that include social barriers within communities, as well as barriers within individual households and health systems. Each of these play a role in women’s ability to access life-saving care. 

Comment: Question: How different is this situation than it is in urban Bangladesh? Are these problems of lack of access and knowledge purely rural? I suspect not, so in that case, how can these findings help those in urban areas? Should this be investigated further? Could you make a recommendation for further research?

Response: Thank you for your comment. This has been addressed already in the limitations section. 

Comment: Your findings will be of interest to rural studies. Please consider using "rural" as a keyword.

Response: Thank you - we’ve added this as a keyword.

---

## [Decision Letter · Decision Letter 2]

13 Nov 2020

PONE-D-19-24100R2

Pathways to service access for pre-eclampsia and eclampsia in rural Bangladesh: exploring women’s care-seeking

PLOS ONE

Dear Dr. Dempsey,

Thank you for submitting your manuscript to PLOS ONE. After careful consideration, we feel that it has merit but does not fully meet PLOS ONE’s publication criteria as it currently stands. Therefore, we invite you to submit a revised version of the manuscript that addresses the points raised during the review process.

ACADEMIC EDITOR:  

While the manuscript has been greatly improved, there remains some concerns regarding grammar and a substantive methodological issue noted by one reviewer. To be considered for publication (very close) the authors need to address these last concerns.. Resubmission is strongly encouraged.

We look forward to receiving your revised manuscript.

Kind regards,

Joseph Telfair, DrPH, MSW, MPH

Academic Editor

PLOS ONE

Reviewers' comments:

Reviewer's Responses to Questions

**Comments to the Author**

1. If the authors have adequately addressed your comments raised in a previous round of review and you feel that this manuscript is now acceptable for publication, you may indicate that here to bypass the “Comments to the Author” section, enter your conflict of interest statement in the “Confidential to Editor” section, and submit your "Accept" recommendation.

Reviewer #1: All comments have been addressed

Reviewer #2: (No Response)

2. Is the manuscript technically sound, and do the data support the conclusions?

Reviewer #1: Yes

Reviewer #2: Yes

3. Has the statistical analysis been performed appropriately and rigorously? 

Reviewer #1: N/A

Reviewer #2: Yes

4. Have the authors made all data underlying the findings in their manuscript fully available?

Reviewer #1: Yes

Reviewer #2: Yes

5. Is the manuscript presented in an intelligible fashion and written in standard English?

Reviewer #1: Yes

Reviewer #2: No

6. Review Comments to the Author

Reviewer #1: (No Response)

Reviewer #2: Please note specific typographical issues in the uploaded document. In particular, the references need to be standardized. I have one further substantive question: I notice that in your interview and discussion group questions you ask about the mother-in-law's influence, but not the mother's. I suspect that this is according to Eastern traditions, where the wife goes to live with the husband's family. If this is the case, can you please note that interviewees were not asked about the mother's influence, according to cultural family forms where the wife lives with the husband's family? Western audiences may need assistance to recognize the importance of the mother-in-law relationship.

7. PLOS authors have the option to publish the peer review history of their article (what does this mean?). If published, this will include your full peer review and any attached files.

Reviewer #1: **Yes: **Florian R. Schroeck

Reviewer #2: **Yes: **Silvia L Vilches

---

## [Author Response · Author response to Decision Letter 2]

27 Nov 2020

Dr. Joerg Hever

Editor in Chief, PLOS ONE

1160 Battery Street 

Koshland Building East, Suite 225

San Francisco, CA 94111

November 13, 2020

Dear Dr. Hever, 

Thank you for the review of the manuscript entitled, “Pathways to service access for pre-eclampsia and eclampsia in rural Bangladesh: exploring women’s care-seeking.” We hope this revised manuscript is acceptable for publication in PLOS ONE. 

We are grateful for the reviewers’ comments and are glad they feel the presentation of pre-eclampsia and eclampsia survivors’ experiences shared in this manuscript is clear and powerful. The focus on these women’s personal stories of accessing maternal health services while experiencing pregnancy-related complications is imperative for improving community perceptions of formal health systems and increasing access to lifesaving care. We welcome the chance to continue improving our manuscript. Please find below detailed responses to the reviewer’s comments and descriptions of corresponding revisions made in the resubmitted manuscript. 

This manuscript comes from original work and is not published or under consideration for publication elsewhere. The authors have no competing interests, and we have approved the manuscript for submission

We thank you for your consideration and look forward to hearing from you.

Sincerely,

Amy Dempsey, MA

Population Council

4301 Connecticut Ave NW, Suite 280

Washington, DC 20008

Reviewer Comment: Please note specific typographical issues in the uploaded document. In particular, the references need to be standardized. I have one further substantive question: I notice that in your interview and discussion group questions you ask about the mother-in-law's influence, but not the mother's. I suspect that this is according to Eastern traditions, where the wife goes to live with the husband's family. If this is the case, can you please note that interviewees were not asked about the mother's influence, according to cultural family forms where the wife lives with the husband's family? Western audiences may need assistance to recognize the importance of the mother-in-law relationship.

Response: It seems that the typos were in the quotes from the IDIs and the FGDs. We had previously decided to leave those unedited to ensure the “voice” of the speaker remained. However, once receiving this comment, we have gone through the text and edited for better English. 

Re: the references, authors have gone through the reference list and reformatted it. 

The authors have added a sentence on page 12 to say: After marriage, it is common for a woman to move into her husband’s parents’ home, and as a result, mothers-in-law also influence women’s health-seeking practices.

---

## [Decision Letter · Decision Letter 3]

30 Dec 2020

Pathways to service access for pre-eclampsia and eclampsia in rural Bangladesh: exploring women’s care-seeking

PONE-D-19-24100R3

Dear Dr. Dempsey,

We’re pleased to inform you that your manuscript has been judged scientifically suitable for publication and will be formally accepted for publication once it meets all outstanding technical requirements.

Kind regards,

Joseph Telfair, DrPH, MSW, MPH

Academic Editor

PLOS ONE

Additional Editor Comments (optional):

Reviewers' comments:

Reviewer's Responses to Questions

**Comments to the Author**

1. If the authors have adequately addressed your comments raised in a previous round of review and you feel that this manuscript is now acceptable for publication, you may indicate that here to bypass the “Comments to the Author” section, enter your conflict of interest statement in the “Confidential to Editor” section, and submit your "Accept" recommendation.

Reviewer #2: All comments have been addressed

2. Is the manuscript technically sound, and do the data support the conclusions?

Reviewer #2: Yes

3. Has the statistical analysis been performed appropriately and rigorously? 

Reviewer #2: Yes

4. Have the authors made all data underlying the findings in their manuscript fully available?

Reviewer #2: No

5. Is the manuscript presented in an intelligible fashion and written in standard English?

Reviewer #2: Yes

6. Review Comments to the Author

Reviewer #2: There are three minor typographical errors. In references #7, #14, and #17 there should be a capitalization after the colon in the title. For example, in Salazar et al, it should be " .... : A multi-level...." not ".... : a multilevel...."

7. PLOS authors have the option to publish the peer review history of their article (what does this mean?). If published, this will include your full peer review and any attached files.

Reviewer #2: **Yes: **Silvia L. Vilches

---

## [Editor Report · Acceptance letter]

7 Jan 2021

PONE-D-19-24100R3 

Pathways to service access for pre-eclampsia and eclampsia in rural Bangladesh: exploring women’s care-seeking 

Dear Dr. Dempsey:

I'm pleased to inform you that your manuscript has been deemed suitable for publication in PLOS ONE. Congratulations! Your manuscript is now with our production department. 

Kind regards, 

on behalf of

Dr. Joseph Telfair 

Academic Editor

PLOS ONE